# Malaria Detection Accelerated: Combing a High-Throughput NanoZoomer Platform with a ParasiteMacro Algorithm

**DOI:** 10.3390/pathogens11101182

**Published:** 2022-10-14

**Authors:** Shoaib Ashraf, Areeba Khalid, Arend L. de Vos, Yanfang Feng, Petra Rohrbach, Tayyaba Hasan

**Affiliations:** 1Wellman Center for Photomedicine, Massachusetts General Hospital, Harvard Medical School, 40 Blossom Street, Boston, MA 02114, USA; 2Department of Animal Science, McGill University, Sainte-Anne-de-Bellevue, QC H9X3V9, Canada; 3Department of Computer Science, Mathematics Adelphi University, Garden City, NY 11530, USA; 4Department of Biomedical Engineering, Tufts University, Medford, OR 02155, USA; 5Swammerdam Institute of Life Sciences, University of Amsterdam, Science Park 904, 1098 XH Amsterdam, The Netherlands; 6Institute of Parasitology, McGill University, Sainte-Anne-de-Bellevue, QC H9X3V9, Canada; 7Health Sciences and Technology, Massachusetts Institute of Technology, Cambridge, MA 02139, USA

**Keywords:** NanoZoomer, ParasiteMacro, algorithm, microscopy, malaria, *Plasmodium falciparum*, parasite, Giemsa-staining, parasitemia

## Abstract

Eradication of malaria, a mosquito-borne parasitic disease that hijacks human red blood cells, is a global priority. Microscopy remains the gold standard hallmark for diagnosis and estimation of parasitemia for malaria, to date. However, this approach is time-consuming and requires much expertise especially in malaria-endemic countries or in areas with low-density malaria infection. Thus, there is a need for accurate malaria diagnosis/parasitemia estimation with standardized, fast, and more reliable methods. To this end, we performed a proof-of-concept study using the automated imaging (NanoZoomer) platform to detect the malarial parasite in infected blood. The approach can be used as a steppingstone for malaria diagnosis and parasitemia estimation. Additionally, we created an algorithm (ParasiteMacro) compatible with free online imaging software (ImageJ) that can be used with low magnification objectives (e.g., 5×, 10×, and 20×) both in the NanoZoomer and routine microscope. The novel approach to estimate malarial parasitemia based on modern technologies compared to manual light microscopy demonstrated 100% sensitivity, 87% specificity, a 100% negative predictive value (NPV) and a 93% positive predictive value (PPV). The manual and automated malaria counts showed a good Pearson correlation for low- (R^2^ = 0.9377, r = 0.9683 and *p* < 0.0001) as well as high- parasitemia (R^2^ = 0.8170, r = 0.9044 and *p* < 0.0001) with low estimation errors. Our robust strategy that identifies and quantifies malaria can play a pivotal role in disease control strategies.

## 1. Introduction

Malaria is one of the deadliest parasitic infections among humans, especially in tropical regions. Symptoms of the disease include fever, vomiting, headaches, and in severe cases, may result in death [1]. Although it is associated with high morbidity and mortality rates, a decrease in its incidence, including both symptomatic as well as asymptomatic cases have been reported since 2000 [2]. However, according to the last malaria WHO report, issued in 2021, due to a disruption of the global malaria response and services because of the COVID-19 pandemic there were 14 million more malaria cases and 69,000 more deaths in 2020 compared to 2019. Thus, in 2020, 241 million malaria cases with 627,000 malaria associated deaths were reported worldwide [3]. The lifecycle of malaria starts when a *Plasmodium*-infected female *Anopheles* mosquito takes a blood meal from the host, depositing sporozoites into the skin and bloodstream which then migrate to the liver and transform into merozoites. Subsequently, the merozoites are released which invade red blood cells (RBCs), and transform into ring, then trophozoite and finally schizont stages. The schizont then egress from the cell, releasing merozoites that invade new uninfected RBCs, reinitiating the erythrocytic cycle [1].

One of the factors that has contributed towards the persistence of malaria is the scarcity for routine use of automated standardized/optimized diagnostic tools that allow for accurate detection and parasite density estimation in asymptomatic individuals [4,5,6]. Manual methods are routinely used to detect malaria, but the results may differ due to many variables such as human error, different types of strains, etc. To reduce such variation, the scientific community is exploring different options to remove noise in the malaria detection and parasite density estimation process. The emergence of antimalarial drug-resistant strains has further emphasized the importance of accurate malaria diagnosis [7,8]. Parasite density serves as a surrogate for the severity of malaria and monitoring parasitemia levels is important for proper treatment monitoring following successful antimalarial therapy. However, malarial parasitemia/density estimation requires the periodic collection of blood smears until the parasite is no longer measurable in blood circulation [9,10], and this exercise is exhausting.

*Plasmodium* spp. are currently identified with light microscopy, rapid diagnostic tests, and molecular detection assays (e.g., PCR), each having their own drawbacks. The standard laboratory diagnosis procedure involves identifying the *Plasmodium* spp. through light microscopy, which remains the gold standard method. However, this approach is time-consuming, cumbersome, and error-prone with different caveats at each step starting from sample preparation to visualization of the parasites. Light microscopy has certain limitations due to a poor-quality control system with the possibility of misdiagnosis especially in cases with low parasitemia or mixed infections [11,12,13,14]. For diagnosis using light microscopy, both thick and thin blood films are prepared. The thick film offers the highest possible malaria screening sensitivity while the thin film offers the best method for identification of parasite species [15]. The primary limitation associated with light microscopy is the varying detection limit of 5–100 parasites/μL [16,17], and human error. The use of rapid diagnostic tests are based on antigen-antibody interactions, and show low sensitivity (low detection limit of 100 parasites/μL) as well as low diagnostic accuracy [17]. Molecular assays, such as PCR, are increasingly being used to detect the presence of *Plasmodium* species [18]. The sensitivity of the PCR based assays is high and ranges from 1-5 parasites/μL [19,20]. However, there are limitations to the use of PCR as it requires expensive equipment, solutions, enzymes, etc. that are not readily obtained in remote areas of Africa. Some items require refrigeration or freezing that is very difficult or impossible to acquire, electrical power is not reliable everywhere, and power outages occur. Furthermore, molecular techniques require experienced personnel, well-trained staff, and a sound quality assurance system [21]. Moreover, they cannot differentiate between new and old infections.

The NanoZoomer HT Scan system 2.0 (Hamamatsu, Shizuoka, Japan) is a slide scanner that is suitable for high-throughput applications with high sensitivity, and high resolution. The salient features of the NanoZoomer include, scanning capability of standard-size (26 mm × 76 mm) slides, the slide loader can handle upto 210 slides at a time/per batch, it is fully automatic without the need of an attendant, wide range of software solutions and Z-stack feature. The system has the capacity to scan the area of 1 cm × 1 cm in 1 min [22]. The system has two scanning modes (at 20× and 40× objectives) and can be directly connected to the scanner control for direct applications which can be leveraged to load, adjust, and focus the slides, etc. The system does not require oil immersion lenses (1000× magnification) and the images can be magnified by the built-in software. In the past other automated platforms for computerized malaria diagnosis have also been tested, such as the Parasight desktop platform which is manufactured by Sight Diagnostics [23]. Nevertheless, the focus of the current study was to evaluate the feasibility of using the NanoZoomer platform and the ParasiteMacro algorithm for detection of malaria.

Literature search shows that the diagnostic accuracy of malaria by any method relies on many factors and can be adversely affected by the inter-examiner variability, observer bias, as well as human errors imposed by large-scale diagnoses in resource-constrained areas of endemic regions [24]. To overcome these caveats, many researchers have been attracted to the prospects of automated malaria detection incorporated into machine and deep learning platforms with their apparent benefits. Most of these studies have used systems that are not suitable for high endemic areas, where the number of malaria cases are high, or low transmission areas, where parasitemia levels are low [6,25,26,27]. Therefore, the objective of this study was to use an automated platform, incorporate currently used methods with modern technologies that will make malaria diagnosis easy, quick, and accurate. 

## 2. Methods

All reagents used in the study were research grade. A+ blood was purchased from Bio IVT (Garden City, NY, USA). Parasite strains (HB3 and 7G8) were obtained from BEI Resources (Manassas, VA, USA) [28].

### 2.1. Parasite Culture

Parasites (*P. falciparum*) were cultivated at 1% hematocrit in RBCs in Roswell Park Memorial Institute 1640 medium (RPMI 1640) [29] supplemented with Albumax II at 37 °C, and 5% CO_2_-5% O_2_. Media was changed every 6 h and growth of the parasites were monitored by Giemsa-staining [30]. Both HB3 (drug sensitive) and 7G8 (drug resistant) strains were grown in parallel. Following establishment of cultures, HB3 was maintained at high parasitemia, while 7G8 was maintained at low parasitemia, i.e., 4–8% versus 0.1–1%, respectively, for testing purposes. Parasitemia was estimated by manual counting of Giemsa-stained infected RBC (iRBC) malaria cultures using a light microscope with a 100× oil immersion objective (Axiophot, Zeiss, Aalen, Germany). Parasitemia was obtained by counting more than 2000 RBCs or 100 fields of view [31] of Giemsa-stained thin blood smears.

### 2.2. NanoZoomer Imaging

For automated counting, malaria thin blood smear slides were digitized using the NanoZoomer HT Scan system 2.0 (Hamamatsu, Japan) (Figure 1). The imaging system consists of three TDI-CCD sensors. The platform can scan up to 210 slides automatically and requires approximately 1 min to scan an area of 10 mm × 10 mm. Standard glass slides were used with 20× (0.46 µm/pixel) objectives [32]. A direct connection to the scanner control was established via the application programming interface. This provided several control measures, such as: upload slide, load slide, initiate focusing, and start scan. This allowed for a bidirectional communication by obtaining live scan information during the scanning process. A total of 60 Giemsa-stained smears/slides were loaded on the NanoZoomer platform, which consisted of 40 slides with varying parasitemia levels and 20 control slides (non-infected RBCs). Images were taken and viewed with NDP viewer (Hamamatsu, Japan) [33]. Scan software and the developed algorithms were run on a personal computer.

### 2.3. Proof-of-Principle Testing—Malaria Counter (ParasiteMacro)

#### 2.3.1. ImageJ Setup

Giemsa-stained thin blood smear slides were collected and photographed at the Pathology Core, Wellman Center for Photomedicine, Massachusetts General Hospital using the NanoZoomer (Hamamatsu, Japan) [34] and light microscope (Axiophot, Zeiss, Germany). To analyze our dataset structure, we took images of RBCs that were infected and uninfected. The code we developed was used to further detail the total number of images. Figure 1 and Figure 2 show the balanced dataset of images from which the data frame was built for further datasets. ImageJ (version 1.41), an open-access Java-based image-processing program, was used for image analysis [35]. The macro first removes the background noise by converting the image into a binary image that causes pixels to either have the value 0 (black) or the maximal value of 255 (white). This is done based on a threshold set at the pixel value of the background. If a pixels’ value is higher than the set threshold it will become black, if it’s lower, than it will become white. The RBCs’ cell membranes have a lower pixel value than the background and will therefore become white, while the background has a higher pixel value and will therefore become black. The resulting image allows for accurate particle discernability and processability during the subsequent selection criteria.

Particle size (area) and the degree of roundness were additional inclusion variables. ImageJ preprocessing and particle analysis commands provided a preliminary particle count, and the software plotted the particle size-frequency distributions. The ImageJ macro was recorded from within the opensource ImageJ software and was enhanced to accurately find and count infected and non-infected RBCs based on their difference in appearance and associated pixel value. Infected RBCs were visible in blue due to the stain (Giemsa) targeting hemozoin crystals present in them. Blue has a lower pixel value compared to uninfected RBCs.

#### 2.3.2. ImageJ Code

Briefly, the code first used the “make binary” function to divide the structures in the image into two-pixel values 0 and 255. Then, the function “set measurements” removed the background by excluding all the high pixel values showing the remaining hemozoin particles. The “analyzed particle” function assigned regions of interest (ROI) to the hemozoin particles, counted by the following “for loop” function. The hemozoin crystals’ ROIs were then tracked in a separate image in two separate stacks, allowing the user to control for accurate crystal inclusion manually.

In the second section of the code, the original image was re-opened, and pixel value thresholds were set to distinguish between infected and non-infected RBCs. A size inclusion threshold was set to reduce included artifacts by the “analyzed particles”. The macro repeated the conversion into a binary mask to find the particles. The “fill holes” component was used to create homogenous particles to aid the “analysis particle” function. The “watershed” function then roughly separated aggregated particles. The “analyze particle” function assigned ROIs to the found particles, and the macro opened the original RGB image with the overlaid ROIs, which allowed for manual graphic interpretation. The “for loop” function counted either red or blue cells based on the set pixel value thresholds. The final portion of the code calculated the hemozoin crystals, the total RBCs and estimated parasitemia in the image and “printed” the values in the results window.

### 2.4. Statistical Analysis

Pearson correlation coefficient was calculated by a two-tailed analysis using Graphpad Prism (9.0).

## 3. Results

### 3.1. In Vitro Malaria Culture and Counting

Both HB3 and 7G8 parasite strains were successfully cultured, after which the parasites were diluted to varying parasitemia levels. The level of parasitemia was estimated by Giemsa-stained thin blood smears that gave parasite levels ranging from 4.8–7.9% for HB3 and 0.15–0.89% for 7G8 strains. To validate the high-throughput NanoZoomer detection system for detecting malaria and evaluating parasitemia, images were taken from slides prepared from the parasite cultures. These images were imported to a computer and the algorithm (ParasiteMacro) was run to estimate parasitemia (Figure 2). A total of 60 samples/slides were analyzed: 20 from cultures with high parasitemia, 20 from cultures with low parasitemia, and 20 from cultures of uninfected RBCs alone serving as a negative control (non-infected RBCs).

### 3.2. Sensitivity, Specificity, Negative and Positive Predictive Values

The results obtained from conventional manual microscopy were considered as true and were compared to the automated malaria detection/parasitemia estimation system (NanoZoomer/ParasiteMacro). The system showed 100% sensitivity, 87% specificity, a 100% negative predictive value (NPV), and a 93% positive predictive value (PPV) (Table 1). The equations [36] used to calculate these parameters are shown below, with abbreviations True positive (T_p_), False positive (F_p_), True negative (T_n_), False negative (F_n_).
Sensitivity% = [T_p_/(T_p_ + F_n_)] × 100(1)
Specificity% = [T_n_/(T_n_ + F_p_)] × 100(2)
Negative predictive value (NPV)% = [T_n_/(T_n_ + F_n_)] × 100(3)
Positive predictive value (PPV)% = [T_p_/(T_p_ + F_p_)] × 100(4)

### 3.3. Comparison of the Automated NanoZoomer ParasiteMacro versus Manual Microscopy

We compared malaria parasitemia of the *P. falciparum* varying infections between the automated versus the manual malaria counts for both high and low parasitemia levels by the scatter plot analysis. The automated and manual counts showed a good Pearson correlation (R^2^ = 0.8170, r = 0.9044 and *p* < 0.0001) for the high as well as the low parasitemia (R^2^ = 0.9377, r = 0.9683 and *p* < 0.0001) (Figure 3).

### 3.4. Automated versus Manual Parasitemia Error Estimation

The automated parasitemia estimates were recorded and compared to the corresponding manually established counterparts for HB3 (Table 2), 7G8 (Table 3) and non-infected RBCs (negative control) (Table 4). In the HB3 strain, error in estimation of parasitemia ranged from 0.1–0.9. In the 7G8 strain, the minimum error recorded was 0.01 and the maximum was 0.16. In the non-infected RBCs, the error was between 0.09–0.17.

## 4. Discussion

In the present study, we have developed a machine learning platform to make malaria detection robust. Our algorithm (ParasiteMacro) is compatible with online public access software using low power objectives in automated (e.g., NanoZoomer) and manual (e.g., routine microscope) imaging systems. We developed a Java code, and its application helps in the fast analysis of malaria. To test our system in this study we used two different *P. falciparum* strains (HB3 and 7G8) at high and low parasitemia, respectively. HB3 is chloroquine sensitive whereas 7G8 is chloroquine resistant. Both strains have different biological characteristics and their inclusion in the study has further strengthened our conclusions on the feasibility of using our novel approach for detection of malaria. To this end, we found our approach to be cumulatively 100% sensitive and the specificity was 87%. The reduced specificity was due to false positives when using the 5× or 10x objectives. This could be due to overstaining of the slides or poor washing after Giemsa stain. The NPV and PPV were calculated as 100% and 93%, respectively.

An expert microscopist takes an average of 15–30 minutes to diagnose malaria and estimate the level of parasitemia in a thin blood smear Giemsa-stained slide [5]. In the proposed automated approach, the NanoZoomer can process 210 slides during the same time, as it scans a 10 mm × 10 mm area in 1 minute [32,33]. This results in an approximate 90% reduction in time to provide a result, or an increase of 10 diagnostic tests per 30 minutes. Moreover, the Institute of Clinical and Laboratory Standards (CLSI) recommends that repeated blood films be collected and analyzed every 6 to 8 h for up to 3 days (if clinically indicated) before malaria is definitively removed from differential diagnosis [37]. Similarly, the CDC suggests a total of 3 assessments before ruling out malaria, while blood smears in non-immune individuals should be analyzed every 12 to 24 h [37]. Hence, this approach will reduce economic burden, noise in the data, user variability, and will make malaria detection/diagnosis much more robust.

Smartphones have been used for malaria detection, especially in resource-poor environments [38,39,40]. For this situation, it is possible to apply the developed Java code to a small portable slide reader that can attach to a smartphone, into which a blood slide is inserted, generating the results. For malaria diagnosis, oil immersion lenses (high optical magnification, up to 1000×) are routinely used. This is a major obstacle as it slows down the process, i.e., the reader needs to view more fields for an accurate diagnosis. Our approach is compatible with low/lower magnification objectives (5×, 10×, 20×, and 40×) and can overcome this obstacle. Various aspects of deep learning in the diagnosis of malaria have been studied [5]. A convolutional neural network was used to discriminate between infected and un-infected cells in thin blood smears. Deep learning does not require the design of handcrafted features, which is one of its most significant advantages. Other authors [41,42] have applied deep learning to cell segmentation which uses convolutional neural networks. Since deep learning is the overarching machine learning technique nowadays, more work is being done concerning cell classification, cell staging, cell segmentation, and other sub-problems in automated malaria diagnosis. Correct malaria detection provides countries with the knowledge required to target populations that will benefit most from malaria services and allows funding agencies to assess the impact of support. There is also a compelling case for public health to extend and strengthen the diagnosis of malaria, which is only possible through proper detection. The new algorithm-based testing technique can increase detection rates which can translate into the reduced likelihood of outbreaks. Due to its suitability for off-lab conditions, this technique has an extra benefit and is suitable for high-throughput and point-of-care (POC) applications.

In this proof-of-concept study, we show that the NanoZoomer, combined with the Java code (ParasiteMacro), can be widely used to not only identify *Plasmodium* species but also estimate *Plasmodium* associated parasitemia levels (which in our case was *P. falciparum*). Our novel approach is easy to use with no steep learning curve. It is recommended that the application of detection and diagnostic devices be focused on the simplicity of an assay to be implemented with minimum infrastructural requirements, such as a reliable supply of electricity in the area [2]. The method can be used to detect *Plasmodium* species with minimum requirements such as an inexpensive microscope and a computer. The new technique considerably improves the sensitivity of malaria diagnosis. Moreover, the method is proposed to be implemented towards a clinical diagnostic accuracy trial to determine the feasibility of use in clinical cases. We thus hypothesize that this robust methodology can play a pivotal role in informed decision making in busy settings. The code will increase the precision of the calculated reservoir size of infections with low- and high-density parasites influencing malaria epidemiology. The identification of 100% positive samples in this study indicates that this method may be an appropriate survey tool to estimate the amount of the asymptomatic low-density infections accurately. This could also improve the efficacy of mass screening and treatment (MSAT) approaches in low incidence settings, given its improved performance. The method described here can be used as an early steppingstone into malaria detection and parasitemia estimation. However, there are some limitations of our study, i.e., the performance of the system was only validated by in house varying parasitemia cultures and no patient samples were analyzed. The quality of the Giemsa-stained blood smears plays a vital role towards getting quality results. We also did not test the system for identifying the different life stages of the parasite. The technique was only compared to manual malaria counts and should be compared with real-time PCR and flowcytometry, etc. The results should be interpreted carefully as other intraerythrocytic parasites, malformed RBCs, cellular components such as WBCs and platelets can also be stained with Giemsa. Finally, our system was not tested to differentiate various *Plasmodium* species (*P. vivax, P. knowlesi, P. cynomolgi, and P. ovale*, etc.); instead, only *P. falciparum* was used as the model organism for malaria. Future studies focusing on comparing the sensitivity and specificity of other malarial detection methods under field conditions should be done using this technology. However, this would be challenging as in field conditions blood smears are often dirty and contaminated, nevertheless the results presented in our study warrant further investigation of the novel platform.

## 5. Conclusions

The novel method described here using a new Java code algorithm (ParasiteMacro) offers a new machine learning approach with a NanoZoomer-automated framework. In specific cases, such as prevalence estimation, reactive case detection, or MSAT, it is now essential to use robust technologies. Employing our machine learning algorithm to automated detection (NanoZoomer) systems of malaria would allow detection in low density and high prevalence areas. Taken together this platform will help in the eradication of malaria. The next steps would be to test this approach on clinical samples and compare it with other methodologies.

## Figures and Tables

**Figure 1 pathogens-11-01182-f001:**
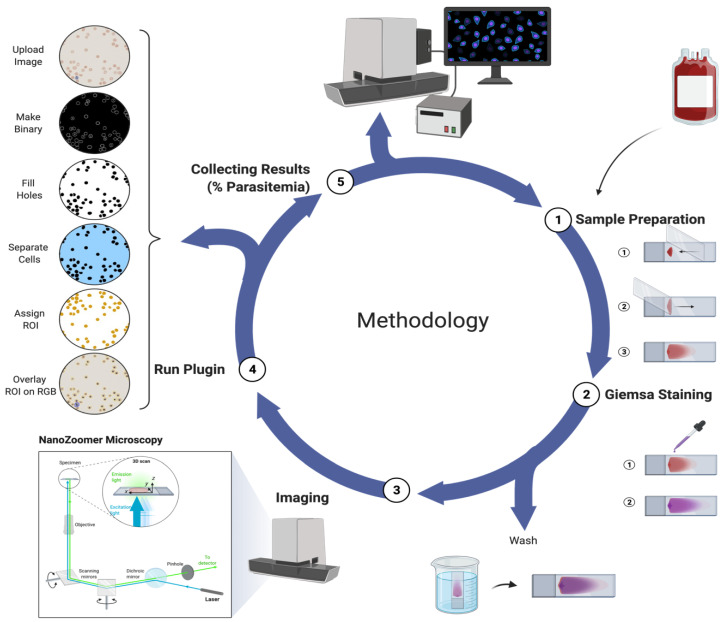
The automated NanoZoomer malaria methodology. (1) Samples are prepared on glass slides, (2) stained with the Giemsa-stain and washed, (3) slides are loaded, and imaged with the NanoZoomer, (4) the ParasiteMacro plugin is run, and (5) the parasitemia of all slides is obtained.

**Figure 2 pathogens-11-01182-f002:**
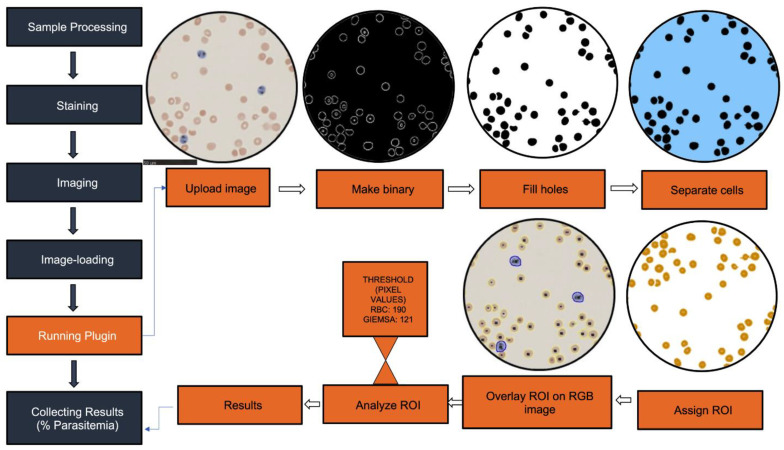
Flow chart and image analysis of automated NanoZoomer malaria counting ParasiteMacro. The acquired images were uploaded to imageJ and the created plugin was run.

**Figure 3 pathogens-11-01182-f003:**
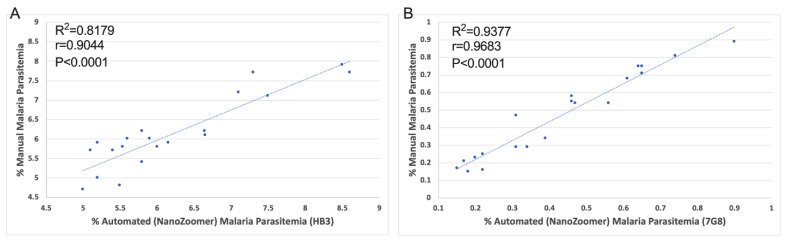
Scatter plots for association of parasitemia determined by automated counting NanoZoomer ParasiteMacro algorithm versus manual counting. (**A**) The HB3 (high parasitemia) strain showed a Pearson correlation coefficient of 0.8179, *p* < 0.0001, (**B**) while the 7G8 strain (low parasitemia) showed a Pearson correlation coefficient of 0.9377, *p* < 0.0001.

**Table 1 pathogens-11-01182-t001:** Malaria diagnosis by the automated NanoZoomer platform using the ParasiteMacro algorithm, true cases were based on the manual microscopy results.

Parameter	Number
Total slides (n)	60
True positives (T_p_)	40
False positives (F_p_)	3
False negatives (F_n_)	0
True negatives (T_n_)	20
Sensitivity%	100
Specificity%	87
Negative predictive value (NPV)%	100
Positive predictive value (PPV)%	93

**Table 2 pathogens-11-01182-t002:** Comparison of automated and manual parasitemia for the HB3 strain (high parasitemia).

Slide Name	Automated Count (NanoZoomer) ParasiteMacro	Manual Count	Error (Automated vs. Manual)
HB3 5 × 1	8.6	7.7	0.9
HB3 5 × 2	8.5	7.9	0.6
HB3 5 × 3	6.64	6.2	0.44
HB3 5 × 4	7.3	7.7	−0.4
HB3 5 × 5	7.5	7.1	0.4
HB3 10 × 1	6.65	6.1	0.55
HB3 10 × 2	6.15	5.9	0.25
HB3 10 × 3	5	4.7	0.3
HB3 10 × 4	6	5.8	0.2
HB3 10 × 5	5.2	5	0.2
HB3 20 × 1	5.5	4.8	0.7
HB3 20 × 2	5.6	6	−0.4
HB3 20 × 3	5.8	5.4	0.4
HB3 20 × 4	5.1	5.7	−0.6
HB3 20 × 5	5.2	5.9	−0.7
HB3 40 × 1	7.1	7.2	0.1
HB3 40 × 2	5.4	5.7	−0.3
HB3 40 × 3	5.54	5.8	−0.26
HB3 40 × 4	5.8	6.2	−0.4
HB3 40 × 5	5.9	6	−0.1

**Table 3 pathogens-11-01182-t003:** Comparison of automated and manual parasitemia for the 7G8 strain (low parasitemia).

Slide Name	Automated Count (NanoZoomer) ParasiteMacro	Manual Count Percentage	Error (Automated vs. Manual)
7G8 5 × 1	0.17	0.21	−0.04
7G8 5 × 2	0.18	0.15	0.03
7G8 5 × 3	0.15	0.17	−0.02
7G8 5 × 4	0.22	0.25	−0.03
7G8 5 × 5	0.2	0.23	−0.03
7G8 10 × 1	0.34	0.29	0.05
7G8 10 × 2	0.56	0.54	0.02
7G8 10 × 3	0.65	0.75	−0.1
7G8 10 × 4	0.31	0.47	−0.16
7G8 10 × 5	0.64	0.75	−0.11
7G8 20 × 1	0.9	0.89	0.01
7G8 20 × 2	0.74	0.81	−0.07
7G8 20 × 3	0.46	0.58	−0.12
7G8 20 × 4	0.47	0.54	−0.07
7G8 20 × 5	0.61	0.68	−0.07
7G8 40 × 1	0.65	0.71	−0.06
7G8 40 × 2	0.22	0.16	0.06
7G8 40 × 3	0.39	0.34	0.05
7G8 40 × 4	0.46	0.55	−0.09
7G8 40 × 5	0.31	0.29	0.02

**Table 4 pathogens-11-01182-t004:** Comparison of automated and manual parasitemia for the non-infected RBCs.

Slide Name	Automated Count (NanoZoomer) ParasiteMacro	Manual Count Percentage	Error (Automated vs. Manual)
RBC 5 × 1	0.17	0	0.17
RBC 5 × 2	0	0	0
RBC 5 × 3	0.09	0	0.09
RBC 5 × 4	0	0	0
RBC 5 × 5	0	0	0
RBC 10 × 1	0	0	0
RBC 10 × 2	0.1	0	0.1
RBC 10 × 3	0	0	0
RBC 10 × 4	0	0	0
RBC 10 × 5	0	0	0
RBC 20 × 1	0	0	0
RBC 20 × 2	0	0	0
RBC 20 × 3	0	0	0
RBC 20 × 4	0	0	0
RBC 20 × 5	0	0	0
RBC 40 × 1	0	0	0
RBC 40 × 2	0	0	0
RBC 40 × 3	0	0	0
RBC 40 × 4	0	0	0
RBC 40 × 5	0	0	0

## Data Availability

The data supporting the study are included in the article.

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
