# Peer review of "Malaria Detection Accelerated: Combing a High-Throughput NanoZoomer Platform with a ParasiteMacro Algorithm"

_pathogens, 2022, doi:10.3390/pathogens11101182_

Round 1

Reviewer 1 Report

Abstract

- exaggerates the pitfalls of microscopy, i.e., lines 18-20, to say this "is cumbersome" and "jeopardizes the malaria management system" is needlessly dismissive.

- line 28, in my view, 87% specificity is not good enough.

Introduction

Line 43, Anopheles must be italicized.

Line 57, this sentence begins, “To reduce such variation…” but the many sources of variation are not mentioned in the previous sentence or the previous paragraph.

Methods

Line 144, more details needed on the thresholding process.

Line 145, what’s the evidence that subtle morphological staining characteristics of P.f. are necessarily non-biological?

If I understand correctly from Results in Tables 2 and 3, the NanoZoomer results were not significantly better than the manual counts. It would appear that the main advantage is the time it takes for manual vs NanoZoomer, an important consideration.

Discussion

Lines 244-251 – The results using a smartphone, Java code, and small portable slide reader to read a single slide should be tested and reported.

This paragraph needs to be revised. The sudden mention of oil immersion as a significant obstacle does not follow logically from the previous sentence.

The limitations are major: this manuscript is restricted in breadth and applicability because it is Afro-centric, focusing only on P. falciparum.

The lack of patient samples for this proof-of-concept is unacceptable.

Parasite life stages also need to be included in this manuscript.

Lines 285-286, the authors are not clear about Giemsa staining quality, that would certainly add to the variation of slides being read.

I found myself curious to better understand how the NanoZoomer interfaces exactly with microscopy – I think the authors need to explain in more detail that the NanoZoomer has lenses, and does not require a mag of 1000x.

Fig. 1. This information (life-cycle) is basic and the figure seems unnecessary.

There is a need for a careful reading of this ms – the language use is not always clear, and sometimes sentences follow one another without a logical connection. The authors should be able to do better.

Author Response

Reviewer 1

Abstract

- exaggerates the pitfalls of microscopy, i.e., lines 18-20, to say this "is cumbersome" and "jeopardizes the malaria management system" is needlessly dismissive.

Response

We thank the reviewer for his valuable comments and have addressed his concerns by modifying the statement as follows:

 “However, this approach is time-consuming and requires much expertise especially in malaria-endemic countries or in areas with low-density malaria infection. Thus, there is a need for ac-curate malaria diagnosis/parasitemia estimation with standardized, fast, and more reliable methods.” Lines 18-20.

- line 28, in my view, 87% specificity is not good enough.

Response

We thank the reviewer for pointing this out and we agree with him i.e., the specificity is not optimum. However, cumulatively our system performs well while demonstrating 100% sensitivity, 87% specificity, a 100% negative predictive value (NPV) and a 93% positive predictive value (PPV). And we know that in the malaria endemic countries false negative results are the main source of transmission of the disease versus the false positives.

Introduction

Line 43, Anopheles must be italicized.

 Response

Thank you, Anopheles has been, italicized now.

Line 57, this sentence begins, “To reduce such variation…” but the many sources of variation are not mentioned in the previous sentence or the previous paragraph.

 Response

We thank the reviewer for pointing this out and have modified the sentence as follows:

“Manual methods are routinely used to detect malaria, but the results may vary due to many variables such as human error, different types of strains etc. To reduce such variation, the scientific community is exploring different options to remove noise in the malaria detection and parasite density estimation process.” Lines 60-64.

Methods

Line 144, more details needed on the thresholding process.

Response

We thank the reviewer for pointing this out and have added the following text in the revised manuscript, lines 212-220.

“The macro first removes the background noise by converting the image into a binary image that causes pixels to either have the value 0 (black) or the maximal value of 255 (white). This is done based on a threshold set at the pixel value of the background. If a pixels’ value is higher than the set threshold it will become black, if it’s lower, than it will become white. The RBCs’ cell membranes have a lower pixel value than the background and will therefore become white, while the background has a higher pixel value and will therefore become black. The resulting image allows for accurate particle discernability and processability during the subsequent selection criteria.

 Particle size (area) and the degree of roundness were additional inclusion variables. ImageJ preprocessing and particle analysis commands provided a preliminary particle count, and the software plotted the particle size-frequency distributions. The ImageJ macro was recorded from within the opensource ImageJ software and was enhanced to accurately find and count infected and non-infected RBCs based on their difference in appearance and associated pixel value. Infected RBCs were visible in blue due to the stain (Giemsa) targeting hemozoin crystals present in them. Blue has a lower pixel value compared to uninfected RBCs.”

Line 145, what’s the evidence that subtle morphological staining characteristics of P.f. are necessarily non-biological?

 Response

We agree with the reviewer and thus we have removed this statement from the revised manuscript on lines 181-182.

If I understand correctly from Results in Tables 2 and 3, the NanoZoomer results were not significantly better than the manual counts. It would appear that the main advantage is the time it takes for manual vs NanoZoomer, an important consideration.

 Response

We thank and agree with the reviewer and would like to emphasize that as we were considering manual counting of Giemsa-stained thin smears as the gold standard, thus, we have compared our results with this technique. However, for the advantage of speed we have added the following sentences in the discussion of the manuscript

“In the proposed automated approach, the NanoZoomer can process 210 slides during the same time, as it scans a 10 mm x 10 mm area in 1 minute [29,30]. This results in an approximate 90% reduction in time to provide a result, or an increase of 10 diagnostic tests per 30 minutes. Moreover, the Institute of Clinical and Laboratory Standards (CLSI) recommends that repeated blood films be collected and analyzed every 6 to 8 h for up to 3 days (if clinically indicated) before malaria is definitively removed from differential diagnosis [31].” Lines 239-260

Discussion

Lines 244-251 – The results using a smartphone, Java code, and small portable slide reader to read a single slide should be tested and reported. 

Response

We thank the reviewer for this suggestion and would like to request that we plan to test these and compare the results with other techniques such as PCR, flowcytometry etc. in a future study. However, at this point as the grant has ended for this project, we do not have more resources to continue the study.

This paragraph needs to be revised. The sudden mention of oil immersion as a significant obstacle does not follow logically from the previous sentence. 

Response

We thank the reviewer for pointing this out and have revised the manuscript as follows

“For malaria diagnosis, oil immersion lenses (high optical magnification, up to 1000x) are routinely used. This is a major obstacle as it slows down the process i.e., the reader needs to view more fields for an accurate diagnosis. Our approach is compatible with low/lower magnification objectives (5x, 10x, 20x, and 40x) and can overcome this obstacle.” Lines 266-269

The limitations are major: this manuscript is restricted in breadth and applicability because it is Afro-centric, focusing only on P. falciparum. The lack of patient samples for this proof-of-concept is unacceptable.  Parasite life stages also need to be included in this manuscript.

Response

We thank the reviewer for pointing this out and we would like to reiterate that we plan to do these experiments in a future study. To this end, this was a preliminary proof-of-principle study where we have tested our set-up i.e., the NanoZoomer, our ParasiteMacro code etc. And we want to report these interesting findings as soon as possible through a credible platform. However, we have now added the following limitations in the manuscript to address the reviewer’s concerns.

““The method described here can be used as an early steppingstone into malaria detection and parasitemia estimation. However, there are some limitations of our study, i.e., the performance of the system was only validated by in house varying parasitemia cultures and no patient samples were analyzed. The quality of the Giemsa-stained blood smears plays a vital role towards getting quality results. We also did not test the system for identifying the different life stages of the parasite. The technique was only compared to manual malaria counts and should be compared with real-time PCR and flowcytometry etc. The results should be interpreted carefully as other intraerythrocytic parasites, malformed RBCs, cellular components such as WBCs and platelets can also be stained with Giemsa. Finally, our system was not tested to differentiate various Plasmodium species (P. vivax, P. knowlesi, P. cynomolgi, and P. ovale etc.); instead, only P. falciparum was used as the model organism for malaria. Future studies focusing on comparing the sensitivity and specificity of other malarial detection methods under field conditions should be done using this technology. However, this would be challenging as in field conditions blood smears are often dirty and contaminated, nevertheless the results presented in our study warrant further investigation of the novel platform.””

Lines 285-286, the authors are not clear about Giemsa staining quality, that would certainly add to the variation of slides being read.

Response

We agree with the reviewer that the quality of the Giemsa staining would add variation in the process that is why we have added a caution in the limitation section as follows

“Furthermore, the quality of the Giemsa-stained blood smears plays a vital role towards getting quality results.” Lines 306-307

I found myself curious to better understand how the NanoZoomer interfaces exactly with microscopy – I think the authors need to explain in more detail that the NanoZoomer has lenses, and does not require a mag of 1000x. 

Response

We thank the reviewer for pointing this out and now we have included a paragraph in the introduction that describes the salient features of the NanoZoomer, which reads as follows

“The NanoZoomer HT Scan system 2.0 (Hamamatsu, Japan) is a slide scanner that is suitable for high-throughput applications with high sensitivity, and high resolution. The salient features of the NanoZoomer include, scanning capability of standard-size (26 mm x 76 mm) slides, the slide loader can handle upto 210 slides at a time/per batch, it is fully automatic without the need of an attendant, wide range of software solutions and Z-stack feature. The system has the capacity to scan the area of 1 cm x 1 cm in 1 min [22]. The system has two scanning modes (at 20x and 40x objectives) and can be directly connected to the scanner control for direct applications which can be leveraged to load, adjust, and focus the slides etc. The system does not require oil immersion lenses (1000x magnification) and the images can be magnified by the built-in software. In the past other automated platforms for computerized malaria diagnosis have also been tested, such as the Parasight desktop platform which is manufactured by Sight Diagnostics [23]. Nevertheless, the focus of the current study was to evaluate the feasibility of using the NanoZoomer platform and the ParasiteMacro algorithm for detection of malaria.” Lines 96-147.

Fig. 1. This information (life-cycle) is basic and the figure seems unnecessary.

Response

We thank the reviewer for pointing this out and have deleted the figure now in the revised manuscript.

There is a need for a careful reading of this ms – the language use is not always clear, and sometimes sentences follow one another without a logical connection. The authors should be able to do better.

Response

We thank the reviewer and have revised the whole manuscript with track changes.

Reviewer 2 Report

The article entitled “Malaria Detection Accelerated: Combing a High-throughput 2 NanoZoomer Platform with a ParasiteMacro Algorithm” presents and describes a novel method for detection of malaria cases appropriate for use in regions with high or low prevalence of the disease. If implemented, such novel approach may be of good help for malaria surveillance and control in the affected areas.

The article is well structured, comprehensive, supplemented with appropriate figures and tables and presents the data sound and clear.

Probably, in the Introduction section at lines 39-42 it may be added that: “… and according to the last malaria WHO report, issued in 2021, due to a disruption of the global malaria response and services because of COVID-19 pandemic there are 14 million more malaria cases and 47 000 more deaths in 2020 compared to 2019 (World malaria report 2021. Geneva: World Health Organization; 2021. Licence: CC BY-NC-SA 3.0 IGO).”

In my opinion the Introduction section should end at line 97. Lines 98 and 101 belong to section Methods.

In the Discussion section, line 265-267 (In this proof-of-concept study, we show that the NanoZoomer, combined with the 265 Java code (ParasiteMacro), can be widely used to not only identify Plasmodium species but 266 also estimate parasitemia levels.), please clarify that the method only detects Plasmodium spp., but do not identify the species, as you state in lines 287-289 (Finally, our system was not tested to differentiate various Plasmodium species; instead only P. falciparum was used as the model organism for malaria.).

My opinion is that the article may be published as is.

Author Response

Reviewer 2

The article entitled “Malaria Detection Accelerated: Combing a High-throughput 2 NanoZoomer Platform with a ParasiteMacro Algorithm” presents and describes a novel method for detection of malaria cases appropriate for use in regions with high or low prevalence of the disease. If implemented, such novel approach may be of good help for malaria surveillance and control in the affected areas. 

The article is well structured, comprehensive, supplemented with appropriate figures and tables and presents the data sound and clear.

Response

We thank the reviewer for these kind comments

Probably, in the Introduction section at lines 39-42 it may be added that: “… and according to the last malaria WHO report, issued in 2021, due to a disruption of the global malaria response and services because of COVID-19 pandemic there are 14 million more malaria cases and 47 000 more deaths in 2020 compared to 2019 (World malaria report 2021. Geneva: World Health Organization; 2021. Licence: CC BY-NC-SA 3.0 IGO).”

Response

We thank the reviewer and have added the following in the revised manuscript

“However, according to the last malaria WHO report, issued in 2021, due to a disruption of the global malaria response and services because of the COVID-19 pandemic there were 14 million more malaria cases and 69,000 more deaths in 2020 compared to 2019. Thus, in 2020 241 million malaria cases with 627,000 malaria associated deaths were reported worldwide [3].” Lines 40-45

In my opinion the Introduction section should end at line 97. Lines 98 and 101 belong to section Methods.

Response

We thank the reviewer and we have deleted lines 110-115 from the revised manuscript

In the Discussion section, line 265-267 (In this proof-of-concept study, we show that the NanoZoomer, combined with the 265 Java code (ParasiteMacro), can be widely used to not only identify Plasmodium species but 266 also estimate parasitemia levels.), please clarify that the method only detects Plasmodium spp., but do not identify the species, as you state in lines 287-289 (Finally, our system was not tested to differentiate various Plasmodium species; instead only P. falciparum was used as the model organism for malaria.).

Response

We thank the reviewer for pointing out and have modified the manuscript as follows

“In this proof-of-concept study, we show that the NanoZoomer, combined with the Java code (ParasiteMacro), can be widely used to not only identify Plasmodium species but also estimate Plasmodium associated parasitemia levels (which in our case was P. falci-parum).” Lines 297-300

My opinion is that the article may be published as is.

Response

We thank the reviewer for accepting our article.

Reviewer 3 Report

1.      The introduction should be improved

2.      Fig 1 should be deleted

3.      Line 54 – “One of the factors that has contributed towards the persistence of malaria is the lack of automated standardized/optimized diagnostic tools” – In my opinion, this should be rephrased as many studies have been conducted on this topic

4.      Line 54 – “To reduce such variation” – Please explain what type of variations you are talking about

5.      In the introduction, it is mentioned that the system is for early detection of malaria but the experiment has not been performed for early detection in patients this line should be deleted

6.      Line 60 – “Parasitemia is one of the indicators of” – needs to be rephrased

7.      Line 93 “Most of these studies have used systems that are not suitable for high endemic areas” – explain why not useful under such conditions

8.      A review of nanoZoomer and other digital platforms should be included in introduction

9.      Reference for stains should be provided

10.   Add references in the materials and methods section (section 2.3)

11.   Along with conventional microscopy real time PCR or flow cytometry should have been included for comparison of results

12.   Delete line 184-186

13.   In table 2 and 3 error doesn’t indicate if the value was more in the manual method or automated system

14.   What was the benefit of including two different stains in the study? This should be discussed.

15.   Samples from patients should also have been included in this comparison

16.   The first paragraph of the discussion may be deleted as it is a repetition of the experimental design  

Author Response

Reviewer 3

The introduction should be improved

  1. Fig 1 should be deleted

Response

We thank the reviewer, figure 1 is now deleted from the manuscript

  1. Line 54 – “One of the factors that has contributed towards the persistence of malaria is the lack of automated standardized/optimized diagnostic tools” – In my opinion, this should be rephrased as many studies have been conducted on this topic

Response

We thank the reviewer for this valuable suggestion, and we have now made following changes in the revised manuscript

“One of the factors that has contributed towards the persistence of malaria is the scarcity for routine use of automated standardized/optimized diagnostic tools that allow for accurate detection and parasite density estimation in asymptomatic individuals [4-6].“ Lines 65-67

  1. Line 54 – “To reduce such variation” – Please explain what type of variations you are talking about

Response

We thank the reviewer for pointing this out and have revised the manuscript as follows

“Manual methods are routinely used to detect malaria, but the results may vary due to many variables such as human error, different types of stains etc. To reduce such vari-ation, the scientific community is exploring different options to remove noise in the malaria detection and parasite density estimation process.” Lines 67-71

  1. In the introduction, it is mentioned that the system is for early detection of malaria but the experiment has not been performed for early detection in patients this line should be deleted

Response

We thank the reviewer for this suggestion, and we have the now deleted “early detection” from the introduction and discussion from the revised manuscript.

  1. Line 60 – “Parasitemia is one of the indicators of” – needs to be rephrased

Response

We thank the reviewer and have now rephrased the sentence as follows

“Parasite density serves as a surrogate for the severity of malaria and monitoring parasitemia levels is important for proper treatment monitoring following successful antimalarial therapy. However malarial parasitemia/density estimation requires the periodic collection of blood smears until the parasite is no longer measurable in blood circulation [9,10] and this exercise is exhausting.” Lines 71-77

  1. Line 93 “Most of these studies have used systems that are not suitable for high endemic areas” – explain why not useful under such conditions

Response

We thank the reviewer and would like to respond as follows: most of the studies that use algorithms etc. have the capacity to read only one or a few slides at a time, whereas, in high endemic areas the number of cases can go upto hundreds per day thus such platforms cannot be implemented in high volume or busy settings.

  1. A review of nanoZoomer and other digital platforms should be included in introduction

Response

We thank the reviewer for pointing this out and now we have included a paragraph in the introduction that describes the salient features of the NanoZoomer and another platform Parasight from Sure Diagnositics, which reads as follows

“The NanoZoomer HT Scan system 2.0 (Hamamatsu, Japan) is a slide scanner that is suitable for high-throughput applications with high sensitivity, and high resolution. The salient features of the NanoZoomer include, scanning capability of standard-size (26 mm x 76 mm) slides, the slide loader can handle upto 210 slides at a time/per batch, it is fully automatic without the need of an attendant, wide range of software solutions and Z-stack feature. The system has the capacity to scan the area of 1 cm x 1 cm in 1 min [22]. The system has two scanning modes (at 20x and 40x objectives) and can be directly connected to the scanner control for direct applications which can be leveraged to load, adjust, and focus the slides etc. The system does not require oil immersion lenses (1000x magnification) and the images can be magnified by the built-in software. In the past other automated platforms for computerized malaria diagnosis have also been tested, such as the Parasight desktop platform which is manufactured by Sight Diagnostics [23]. Nevertheless, the focus of the current study was to evaluate the feasibility of using the NanoZoomer platform and the ParasiteMacro algorithm for detection of malaria.”

  1. Reference for stains should be provided

Response

We thank the reviewer for pointing this out and have added the following reference “Parasite strains (HB3 and 7G8) were obtained from BEI Resources (USA) [26].”

  1. Add references in the materials and methods section (section 2.3)

We thank the reviewer have adding the following eight references in the materials and methods section

“28. Molestina, R.E.; Stedman, T.T. Update on BEI Resources for Parasitology and Arthropod Vector Research. Trends in parasitology 2020, 36, 321-324.

  1. Trager, W.; Jensen, J.B. Cultivation of erythrocytic and exoerythrocytic stages of plasmodia. In Pathology, Vector Studies, and Culture; Elsevier: 1980; pp. 271-319.
  2. Trager, W.; Jensen, J.B. Human malaria parasites in continuous culture. Science 1976, 193, 673-675.
  3. Alves-Junior, E.R.; Gomes, L.T.; Ribatski-Silva, D.; Mendes, C.R.J.; Leal-Santos, F.A.; Simoes, L.R.; Mello, M.B.C.; Fontes, C.J.F. Assumed white blood cell count of 8,000 cells/μL overestimates malaria parasite density in the Brazilian Amazon. PLoS One 2014, 9, e94193.
  4. Rojo, M.G.; García, G.B.; Mateos, C.P.; García, J.G.; Vicente, M.C. Critical comparison of 31 commercially available digital slide systems in pathology. International journal of surgical pathology 2006, 14, 285-305.
  5. Lahrmann, B.; Valous, N.A.; Eisenmann, U.; Wentzensen, N.; Grabe, N. Semantic focusing allows fully automated single-layer slide scanning of cervical cytology slides. PloS one 2013, 8, e61441.
  6. Mallidi, S.; Watanabe, K.; Timerman, D.; Schoenfeld, D.; Hasan, T. Prediction of tumor recurrence and therapy monitoring using ultrasound-guided photoacoustic imaging. Theranostics 2015, 5, 289.
  7. Collins, T.J. ImageJ for microscopy. Biotechniques 2007, 43, S25-S30.

  1. Along with conventional microscopy real time PCR or flow cytometry should have been included for comparison of results

Response

We thank the reviewer for pointing this out and like to request that due to lack of funding we had to stop the experiments and thus have included this information in the limitations as follows “The method described here can be used as an early steppingstone into malaria detection and parasitemia estimation. However, there are some limitations of our study, i.e., the performance of the system was only validated by in house varying parasitemia cultures and no patient samples were analyzed. The quality of the Giemsa-stained blood smears plays a vital role towards getting quality results. We also did not test the system for identifying the different life stages of the parasite. The technique was only compared to manual malaria counts and should be compared with real-time PCR and flowcytometry etc. The results should be interpreted carefully as other intraerythrocytic parasites, malformed RBCs, cellular components such as WBCs and platelets can also be stained with Giemsa. Finally, our system was not tested to differentiate various Plasmodium species (P. vivax, P. knowlesi, P. cynomolgi, and P. ovale etc.); instead, only P. falciparum was used as the model organism for malaria. Future studies focusing on comparing the sensitivity and specificity of other malarial detection methods under field conditions should be done using this technology. However, this would be challenging as in field conditions blood smears are often dirty and contaminated, nevertheless the results presented in our study warrant further investigation of the novel platform.”

  1. Delete line 184-186

Response

We have now deleted many redundant and unnecessary lines throughout the manuscript.

  1. In table 2 and 3 error doesn’t indicate if the value was more in the manual method or automated system

Response

We thank the reviewer for pointing out and we have now modified the table headings and the content i.e., have also included that the error was automated versus manual counts.

  1. What was the benefit of including two different stains in the study? This should be discussed.

Response

We thank the reviewer for pointing this out and we have now included the following information in the revised manuscript which reads as “To test our system in this study we used two different P. falciparum strains (HB3 and 7G8) at high and low parasitemia, respectively. HB3 is chloroquine sensitive whereas 7G8 is chloroquine resistant. Both strains have different biological characteristics and their in-clusion in the study has further strengthened our conclusions on the feasibility of using our novel approach for detection of malaria.” Lines 266-271

  1. Samples from patients should also have been included in this comparison

Response

We thank the reviewer for pointing this limitation and we have now revised the limitations in the manuscript which reads as follows “However, there are some limitations of our study, i.e., the performance of the system was only validated by in house varying parasitemia cultures and no patient samples were analyzed. Furthermore, the quality of the Giemsa-stained blood smears plays a vital role towards getting quality results. We also did not test the system for identifying the dif-ferent life stages of the parasite. Moreover, the technique was only compared to manual malaria counts and should be compared with real-time PCR and flowcytometry etc. Fi-nally, our system was not tested to differentiate various Plasmodium species; instead, only P. falciparum was used as the model organism for malaria. Future studies focusing on comparing the sensitivity and specificity of other malarial detection methods under field conditions should be done using this technology.” Lines 348-358

  1. The first paragraph of the discussion may be deleted as it is a repetition of the experimental design  

Response

We thank the reviewer and agree with them and have thus now deleted the first few lines of the discussion.

Reviewer 4 Report

In this article, the author tried to describe a methodology to screen the Plasmodium parasite slides with high accuracy. This automated system could be a breakthrough and could provide advancement in parasite detection, especially in endemic countries deprived of resources. This article is well-written however had a limited context compared to the problem faced in malaria detection.

I have the following comments regarding this manuscript.

  • WHO released the malaria report in Dec 2021, so the author should update the data on lines 41 to 43.
  • In culture conditions, Plasmodium parasites are cultured with 5% hematocrit however, the author cultivated them with 1% HCT. An explanation is required to support that conditions.
  • The system efficiency was checked on the parasite cultured in isolated RBC. However, in the field, the patient sample contains various other blood cells like eosinophils, neutrophils, basophils, lymphocytes, monocytes, and platelets. The nucleus of these cells is also stained blue-purple and could create a problem in parasite detection with this automated system.
  • To detect other parasites like spirochetes Giemsa stain is used. In that situation, it would be hard to differentiate between infection types.
  • Infection with other Plasmodium parasites like P. vivax infecting reticulocytes (immature RBC) is hard to detect. In blood samples, reticulocytes number are very low that's why P. vivax causes severe malaria at a very low parasitemia. In this situation, an automated system could miss the presence of the parasite in low parasitemia patient samples. In P. vivax, only a single parasite propagates in an RBC whereas multiple P. falciparum parasites can grow in an RBC. So this observation helps the microscopist to differentiate between parasite strains. It is difficult to understand how this system could be trained to adopt that specificity.
  • Parasite invasion changes the morphology of the RBC, where its size increases compared to the uninfected RBC. So the system should be well adapted to differentiate between parasite-infected RBC and WBC.
  • In the bloodstream, the malaria parasite forms gametocytes and, their morphology is quite different from the other cells.
  • So far, this system is only checked with P. falciparum and requires further testing with field parasites i.e. P. vivaxP. knowlesiP. cynomolgi, and P. ovale.
  • Overall this system could perform excellently in lab conditions however, in field cases, the blood smears are dirty and contaminated with other blood cells. Malaria patients encounter anemic conditions because of high parasitemia. In all these complications, it seems very hard for an automated system to work with high efficiency and accuracy. These are the limitation we are facing in infectious diseases for so many years where lab methodology is not 

Author Response

Reviewer 4

In this article, the author tried to describe a methodology to screen the Plasmodium parasite slides with high accuracy. This automated system could be a breakthrough and could provide advancement in parasite detection, especially in endemic countries deprived of resources. This article is well-written however had a limited context compared to the problem faced in malaria detection.

Response

We thank the reviewer for their nice suggestions and below we respond to each point raised.

I have the following comments regarding this manuscript.

WHO released the malaria report in Dec 2021, so the author should update the data on lines 41 to 43.

Response

We thank the reviewer and have added the following in the revised manuscript

“Although it is associated with high morbidity and mortality rates, a decrease in its in-cidence, including both symptomatic as well as asymptomatic cases have been reported since 2000 [2]. However, according to the last malaria WHO report, issued in 2021, due to a disruption of the global malaria response and services because of the COVID-19 pan-demic there are 14 million more malaria cases and 69,000 more deaths in 2020 compared to 2019. Thus, in 2020 241 million malaria cases with 627,000 malaria associated deaths were reported worldwide [3].” Lines 38-45

In culture conditions, Plasmodium parasites are cultured with 5% hematocrit however, the author cultivated them with 1% HCT. An explanation is required to support that conditions.

Response

We appreciate the reviewer for highlighting this and would like to respond that different scientists have used HCT values from ranging 1-5% during their studies. And we used 1 % HCT because we were able to establish the malarial infection at these values in our culture conditions. Some examples of other groups who have used such HCT percentages are listed below

Guerra, E.D., Baakdah, F., Georges, E., Bohle, D.S. and Cerruti, M., 2019. What is pure hemozoin? A close look at the surface of the malaria pigment. Journal of Inorganic Biochemistry194, pp.214-222.

Kazarine, A., Baakdah, F., Gopal, A.A., Oyibo, W., Georges, E. and Wiseman, P.W., 2019. Malaria detection by third-harmonic generation image scanning cytometry. Analytical chemistry, 91(3), pp.2216-2223.

The system efficiency was checked on the parasite cultured in isolated RBC. However, in the field, the patient sample contains various other blood cells like eosinophils, neutrophils, basophils, lymphocytes, monocytes, and platelets. The nucleus of these cells is also stained blue-purple and could create a problem in parasite detection with this automated system. To detect other parasites like spirochetes Giemsa stain is used. In that situation, it would be hard to differentiate between infection types. Infection with other Plasmodium parasites like P. vivax infecting reticulocytes (immature RBC) is hard to detect. In blood samples, reticulocytes number are very low that's why P. vivax causes severe malaria at a very low parasitemia. In this situation, an automated system could miss the presence of the parasite in low parasitemia patient samples. In P. vivax, only a single parasite propagates in an RBC whereas multiple P. falciparum parasites can grow in an RBC. So this observation helps the microscopist to differentiate between parasite strains. It is difficult to understand how this system could be trained to adopt that specificity. Parasite invasion changes the morphology of the RBC, where its size increases compared to the uninfected RBC. So the system should be well adapted to differentiate between parasite-infected RBC and WBC. In the bloodstream, the malaria parasite forms gametocytes and, their morphology is quite different from the other cells. So far, this system is only checked with P. falciparum and requires further testing with field parasites i.e. P. vivaxP. knowlesiP. cynomolgi, and P. ovale. Overall this system could perform excellently in lab conditions however, in field cases, the blood smears are dirty and contaminated with other blood cells. Malaria patients encounter anemic conditions because of high parasitemia. In all these complications, it seems very hard for an automated system to work with high efficiency and accuracy. These are the limitation we are facing in infectious diseases for so many years where lab methodology is not 

Response

We agree with the reviewer and have now modified the limitations in the revised manuscript as follows

“The method described here can be used as an early steppingstone into malaria detection and parasitemia estimation. However, there are some limitations of our study, i.e., the performance of the system was only validated by in house varying parasitemia cultures and no patient samples were analyzed. The quality of the Giemsa-stained blood smears plays a vital role towards getting quality results. We also did not test the system for identifying the different life stages of the parasite. The technique was only compared to manual malaria counts and should be compared with real-time PCR and flowcytometry etc. The results should be interpreted carefully as other intraerythrocytic parasites, malformed RBCs, cellular components such as WBCs and platelets can also be stained with Giemsa. Finally, our system was not tested to differentiate various Plasmodium species (P. vivax, P. knowlesi, P. cynomolgi, and P. ovale etc.); instead, only P. falciparum was used as the model organism for malaria. Future studies focusing on comparing the sensitivity and specificity of other malarial detection methods under field conditions should be done using this technology. However, this would be challenging as in field conditions blood smears are often dirty and contaminated, nevertheless the results presented in our study warrant further investigation of the novel platform.”

Round 2

Reviewer 1 Report

The authors have edited the manuscript thoroughly and carefully, much improving it overall.

Congratulations! It was a pleasure to read and think again about the revised ms.

Good luck with the next steps, following up on the proof-of-concept.

Reviewer 4 Report

The author did provide sufficient details and addressed the limitation of this technique.